# Continual Learning via Neural Pruning

Siavash Golkar*
Flatiron Institute
New York University
sgolkar@flatironinstitute.org

Michael Kagan
SLAC National Accelerator Laboratory
makagan@slac.stanford.edu

Kyunghyun Cho
New York University
Facebook AI Research
CIFAR Azrieli Global Scholar
kyunghyun.cho@nyu.edu

## Abstract

Inspired by the modularity and the lifecycle of biological neurons, we introduce Continual Learning via Neural Pruning (CLNP), a new method aimed at lifelong learning in fixed capacity models based on the pruning of neurons of low activity. In this method, an $L_1$ regulator is used to promote the presence of neurons of zero or low activity whose connections to previously active neurons is permanently severed at the end of training. Subsequent tasks are trained using these pruned neurons after reinitialization and cause zero deterioration to the performance of previous tasks. We show empirically that this biologically inspired method leads to state of the art results beating or matching current methods of higher computational complexity.

## 1 Introduction

Continual learning, the ability of models to learn to solve new tasks beyond what has previously been trained, has garnered much attention from the machine learning community in recent years. The main obstacle for effective continual learning is the problem of catastrophic forgetting: machines trained on new problems forget about the tasks that they were previously trained on. There are multiple approaches to this problem, from employing networks with many sub-modules [1, 8, 12] to methods which penalize changing the weights of the network that are deemed important for previous tasks [3, 5, 16]. These approaches either require specialized training schemes or still suffer catastrophic forgetting, albeit at a smaller rate. Furthermore, from a biological perspective, the current fixed capacity approaches generally require the computation of a posterior in weight space which is non-local and hence biologically implausible.

Motivated by the life-cycle of biological neurons [6], we introduce a simple continual learning algorithm for fixed capacity networks which can be trained using standard gradient descent methods and suffers *zero* deterioration on previously learned problems during the training of new tasks. In this method, the only modifications to standard machine learning algorithms are simple and biologically plausible: i.e. a sparsifying $L_1$ regulator and activation threshold based neural pruning. We demonstrate empirically that these modifications to standard practice lead to state of the art performance on standard catastrophic forgetting benchmarks.

## 2 Related work

*Lifelong learning.* Prior work addressing catastrophic forgetting generally fall under two categories. In the first category, the model is comprised of many individual modules at each layer and forgetting is prevented either by routing the data through different modules [1] or by successively adding new modules for each new task [8, 12]. This approach often has the advantage of suffering zero forgetting,

however, the structure of these networks is specialized. In the case of [8, 12], the model is not fixed capacity and in the case of [1] training is done using a tournament selection genetic algorithm. In the second category of approaches to lifelong learning the network structure and training scheme are standard, and forgetting is addressed by penalizing changes of weights which are deemed important [3, 5, 16]. These approaches, generally referred to as weight elasticity methods, have the advantage of simpler training schemes but still suffer catastrophic forgetting, albeit at a smaller rate than unconstrained training.

*Sparsification.* While sparsification is a crucial tool that we use, it is not in itself a focus of this work. For accessibility, we use a simple neuron/filter based sparsification scheme which can be thought of as a single iteration variation of [4].

## 3 Methodology

The core idea of our method is to take advantage of the fact that neural networks are vastly over-parametrized [10] . A manifestation of this over-parametrization is through the practice of sparsification, i.e. the compression of neural network with relatively little loss of performance [2, 4, 13]. As an example, it was shown in [9] show that VGG-16 can be compressed by more than 16 times. In this section we first show that given an activation based sparse network, we can leverage the unused capacity of the model to develop a continual learning scheme which suffers no catastrophic forgetting. We then discuss the idea of graceful forgetting to address the tension between sparsification and model performance in the context of lifelong learning.

In what follows we will discuss sparsity for fully connected layers by looking at the individual neurons. The same argument goes through identically for individual channels of convolutional layers.

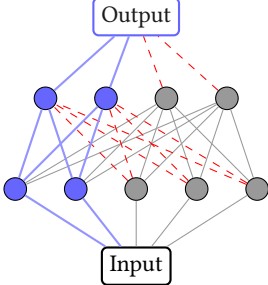

**Figure 1:** The partition of an network with neuronal sparsity into active, inactive and interference parts.

### 3.1 Generalities

Let us assume that we have a trained network which is sparse in the sense that only a subset of the neurons of the network are active.

---
* Work completed while at New York Unviersity.

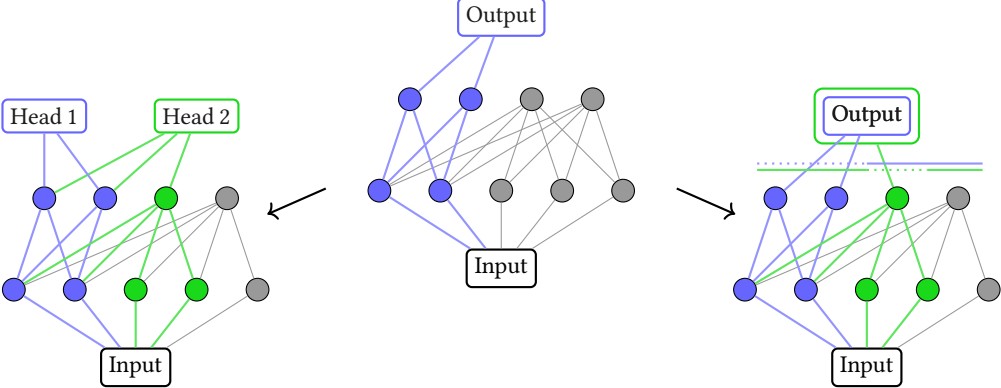

**Figure 2: Middle: A network trained on first task (blue) with neuronal sparsity, where the interference weights have been put to zero. Left, right: the multi-head and single-head expansions on second task (green).**

Networks with this form of sparsity can be thought of as narrow subnetworks embedded inside the original structure. There are many approaches that aim to train such sparse networks with little loss of performance (e.g. [4, 9]). We will discuss our sparsification method in detail in §3.2.

Fig. 1 shows a cartoon of our approach, where we have a network with activation based neuronal sparsity, where the active and inactive neurons are respectively denoted by blue and grey nodes. Based on the connectivity structure, the weights of the network can also be split into three classes. First, denoted in blue in Fig. 1, we have the active weights $W^{\text{act}}$ which connect active nodes to active nodes. Next we have the weights which connect any node to inactive nodes, we call these the free weights $W^{\text{free}}$, denoted in grey. Finally we have the weights which connect the inactive nodes to the active nodes, we call these the interference weights $W^{\text{int}}$, denoted in red dashed lines. A more precise definition of the active and inactive neurons and weights is given in §3.2.

The crux of our approach is the simple observation that if all the interference weights $W^{\text{int}}$ are set to zero, the free weights $W^{\text{free}}$ can be changed arbitrarily without causing any change whatsoever to the output of the network. We can therefore utilize these weights to train new tasks without *any* catastrophic forgetting of the previous tasks.

We can further split the free weights into two groups. First, the weights which connect active nodes to inactive nodes. These are the weights that take advantage of previously learned features and are therefore responsible for transfer learning throughout the network. We also have the weights that connect inactive nodes to inactive nodes. These weights can form completely new pathways to the input and train new features. A simple measure of the amount of transfer learning taking place is the number of new active neurons at each layer after the training of subsequent tasks. Given that an efficient sparse training scheme would not need to relearn the features that are already present in the network, the number of new neurons grown at each stage of training is an indicator of the sufficiency of the already learned features for the purposes of the new task. For example, if the features learned at some layer for previous tasks provide sufficient statistics for purposes of a subsequent task, no new neurons need to be trained at this layer during the training of the subsequent task. We will see more of this point in §4.

*Output architecture.* To fully flesh out a continual learning scheme, we need to specify the connectivity structure of the output nodes. There are two intuitive routes that we can take. In order to train a new task, one option is to use a new output layer (i.e. a new head) while saving the previous output layer. This option, demonstrated in Fig. 2 on the left, is known as the multi-head approach and is standard in continual learning. Because each new output layer comes with its own sets of weights which connect to the final hidden layer neurons, this method is not a fully fixed capacity method. Note that in our approach to continual learning, training a multi-head network with a fully depleted core structure, i.e. a network where are no more free neurons left, is equivalent to final layer transfer learning.

In scenarios where the output layer of the different tasks are structurally compatible, for example when all tasks are classification on the same number of classes, we can use a single-head approach. Demonstrated in Fig. 2 on the right, in this approach we use the same output layer for all tasks, but for each task, we mask out the neurons of the final hidden layer that were trained on other tasks. In the case of Fig. 2, only green nodes in the final hidden layer are connected to the output for the second task and only blue nodes for the first task. This is equivalent to a dynamic partitioning of the final hidden layer into multiple unequal sized parts, one part for each task. In practice this is done using a multiplicative masking operation with a task dependent mask, denoted in Fig. 2 by dashed lines after the final hidden layer. This structure, being truly fixed, is more restrictive to train than its multi-head counterpart. Because of this, single head continual algorithms were not possible previously, and as far as we are aware, CLNP is the first viable such algorithm.

## 3.2 Methodology details

In what follows we will assume that we are using Rectifier Linear Units (ReLU). While we have only tested our methodology with ReLU networks, we expect it to work similarly with other activations.

*Sparsification.* So far in this section we have shown that given a sparse network trained on a number of tasks, we can train the network on new tasks without suffering any catastrophic forgetting. We now discuss the specific scheme that we use to achieve this sparisty, which is similar in spirit to the network trimming approach put forward in Ref. [4].

Our sparsification method is comprised of two parts. First, during the training of each task, we add an $L^1$ weight regulator to promote sparsity and regulate the magnitude of the weights of the network. This is akin to biological energy requirements for synaptic communication. The coefficient of $\alpha$ of this regulator is a hyperparameter of our approach. We can also gain more control over the amount of sparsity in each layer by choosing a different $\alpha$ for different layers. The second part of our sparsification scheme is post-training neuron pruning based on the average activity of each neuron. This step is the analogue of long term depression of synaptic connections between neurons without correlated activities. Subsequently at the beginning of training a new task, the connections of these pruned neurons are reinitialized in a manner reminiscent of the lifecycle of biological neurons [6].

Note that most efficient sparsification algorithms include a third part which involves adjusting the surviving weights of the network after pruning. This step is referred to as fine-tuning and is done by retraining the network for a few epochs while only updating the weights which survive sparsification. This causes the model to regain some of its lost performance because of pruning. To achieve a yet higher level of sparsity, one can iterate the pruning and fine-tuning steps multiple times. For simplicity, unless otherwise specified, we only perform one iteration of pruning without the fine tuning step.

*Partitioning the network.* In §3.1, we split the network into active and inactive parts which we define as follows. Given network $N$, comprised of $L$ layers, we denote the neurons of each layer as $N_l$ with $l = 1 \cdots L$. Let us also assume that the network $N$ has been trained on dataset $S$. In order to find the active and inactive neurons of the network, we compute the average activity over the entire dataset $S$ for each individual neuron. In a network with ReLU activations, we identify the active neurons $N_l^{\text{act}}$, i.e. the blue nodes in Fig. 2, as those whose average activation exceeds some threshold parameter $\theta$: $N_l^{\text{act}} = \{N_l \mid \mathbb{E}_S(N_l) > \theta\}$. The inactive neurons are taken as the complement $N_l^{\text{inact}} = N_l \setminus N_l^{\text{act}}$. The threshold value $\theta$ is a post-training hyperparameter of our approach. Similar to the $L^1$ weight regulator hyperparameter $\alpha$, $\theta$ can take different values for the different layers. Furthermore, if $\theta = 0$, $N_l^{\text{inact}}$ would be given by the neurons in the network which are completely dead and the function being computed by the network is entirely captured in $N_l^{\text{act}}$. We can therefore view $N_l^{\text{act}}$ as a compression of the network into a sub-network of smaller width. Based on their connectivity structure, the weights of each layer are again divided into active, free and interference parts, respectively corresponding to the blue, grey and red lines in Fig. 2.

*Graceful forgetting.* While sparsity is crucial in our approach for the training of later tasks, care needs to be taken so as not to overly sparsify and thereby reduce the model's performance. In practice, model sparsity has a similar relationship with generalization as other regularization schemes. As sparsity increases, initially the generalization performance of the model improves. However, as we push our sparsity knobs (i.e. the $L^1$ regulator $\alpha$ and activity threshold $\theta$) higher, eventually both training and validation accuracy will suffer and the network fails to fit the data properly. This means that in choosing these hyperparameters, we have to make a compromise between model performance and remaining network capacity for future tasks.

This brings us to a subject which is often overlooked in lifelong learning literature generally referred to as graceful forgetting. This is

the general notion that it would be preferable to sacrifice some accuracy in a controlled manner, if it reduces future catastrophic forgetting of this task and also helps in the training of subsequent tasks. We believe any successful fixed capacity continual learning algorithm needs to implement some form of graceful forgetting scheme. In our approach, graceful forgetting is implemented through the sparsity vs. performance compromise. In other words, after the training of each task, we sparsify the model up to some acceptable level of performance loss (given by a margin parameter $m$) in a controlled manner. We then move on to subsequent tasks knowing that the model no longer suffers any further deterioration from training future tasks. This has to be contrasted with other weight elasticity approaches which use soft constraints on the weights of the network and cannot guarantee future performance of previously trained tasks.

The choice of sparsity hyperparameters is made based on this incarnation of graceful forgetting as follows. We scan over a range of hyperparameters ($\alpha$, the $L^1$ weight regulator and $\xi$, the learning rate) using grid search and note the value of the best validation accuracy across all hyperparameters. We then pick the models which achieve validation accuracy within a margin of $m\%$ of this best validation accuracy. The margin parameter $m$ controls how much we are willing to compromise on accuracy to regain capacity and in experiments we take it to be generally in the range of 0.05% to 2% depending on the task. We sparsify the picked models using the highest activation threshold $\theta$ such that the model remains within this margin of the best validation accuracy. We finally pick the hyperparameters which give the highest sparsity among these models. In this way, we efficiently find the optimal hyperparameters $\alpha^*(m)$, $\theta^*(m)$ and $\xi^*(m)$ which afford the highest sparsity model with validation accuracy within $m\%$ of the highest validation accuracy among all hyperparameters.

After pruning away the unused weights and neurons of the model with the hyperparameters chosen as above, we report the test accuracy of the sparsified network. This algorithm for training and hyperparameter grid search does not incur any significant additional computational burden over standard practice. The hyperparameter search is performed in standard fashion, and the additional steps of selecting networks within the acceptable margin, scanning the threshold, and selecting the highest sparsity network only require evaluation and do not include any additional network training.

## 4  Experiments

*Permuted MNIST.* In this experiment, we look at the performance of our approach on ten tasks derived from the MNIST dataset via ten random permutations of the pixels . To compare with previous work, we start with the same structure and hyperparameters as in Ref. [16]: a multi-head MLP architecture with two hidden layers, each with 2000 neurons and ReLU activation and a softmax multi-class cross-entropy loss trained with Adam optimizer and batch size 256. In order to make the task more challenging we look at two variations of this structure: For the first variation, we employ only a single-head to demonstrate the viability of our single-head approach. For the second variation we use layers of width 100 instead of 2000.

For the first network variation (wide single-head structure), we do a search over the hyperparameters on the first task using a heldout validation set, just as in Ref. [16]. For the remaining tasks, we settle on learning rate of 0.002 and $L^1$ weight regularization

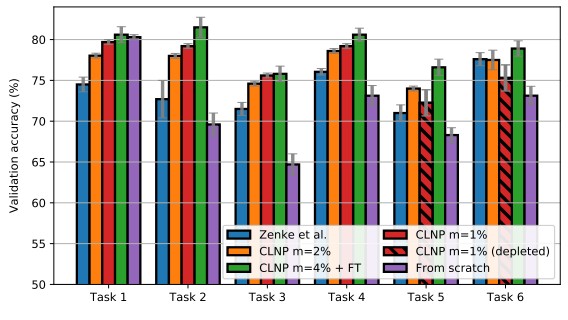

(a) Validation accuracy comparison

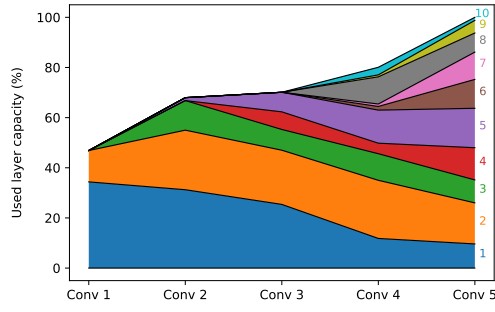

(b) Average network capacity usage per task

Figure 3: CIFAR-10 and split CIFAR-100 results on multi-head network.

$\alpha = 10^{-7}, 10^{-5}, 10^{-6}$ respectively for the first, second and final layers. Finally, when sparsifying after training each task, we allow for graceful forgetting with a small margin of $m = 0.05\%$. With test error within 0.05% of single task SGD training, CLNP virtually eliminates catastrophic forgetting and achieves an average accuracy of $98.42 \pm 0.04$ which is just shy of the single task performance of $98.48 \pm 0.05$ (mean + STD over 5 iterations of the experiment).

In the second variation of the network (narrow multi-head structure), perform sparsification with 2 iterations of fine-tuning and specifically choose a graceful forgetting margin ($m = 3\%$) such that the network is saturated (runs out of free neurons) after all 10 tasks have been trained. In this case, our method attains an average of 95.8% over 10 tasks. In both variations of the network, our results achieve state of the art performance on networks with comparable size, matching or exceeding prior methods of much higher conceptual and computational complexity (e.g. [11]). For an exhaustive comparison of these results with previous methods see Tab. 2 in [14].

*Split CIFAR-10/CIFAR-100.* In this experiment, we train an image classifier sequentially, first on CIFAR-10 (task 1) and then on CIFAR-100 split into 10 different tasks, each with 10 classes (tasks 2-11). We employ the same multi-head network used in Ref. [16], and we use two different training schemes comprised of maximum graceful forgetting of $m = 1\%$ and $m = 2\%$. The validation accuracy of the 6 tasks after training them sequentially is shown in Fig. 3a. We see that we again achieve state of the art performance. The more ambitious $m = 1\%$ scheme (which only allowed for a graceful forgetting of less than 1%) runs out of capacity after the fourth task is trained. We notice that after the model capacity is depleted (tasks 5 and 6 denoted with red dashed lines), the performance of the $m = 1\%$ scheme plummets, showing the necessity for unused neurons for the performance of the network. The more moderate forgetting scheme $m = 2\%$ (denoted in orange), however, maintains high performance throughout all tasks and does not run out of capacity until final task is trained.

We repeated the experiment with a graceful forgetting of $m = 4\%$ but this time followed by fine-tuning, i.e. retraining of the remaining weights after pruning. The results of this method are given in Fig. 3a in green. We see that here there is virtually no catastrophic forgetting on the first task (the model performs even better after pruning and retraining as has been reported in previous sparsity literature [4, 7]). The remaining tasks also get a significant boost from this improved

sparsification method. This is a simple demonstration of the potential of sparsification based continual learning methods given more advanced sparsification schemes.

We also use a wider single-head network for comparison. In Fig. 3b, we can see the number of new channels learned at each layer for each consecutive task. Of note, the first convolutional layer trains new channels only for tasks 1 and 2. The second and third convolutional layers, grow new channels up to task 3 and task 5 respectively. The fourth layer keeps training new channels up to the last task. The fact that the first layer grows no new channels after the second task implies that the features learned during the training of the first two tasks are eemed sufficient for the training of the subsequent tasks. The fact that this sufficiency happens after training more tasks for layers 2 and 3 is a verification of the fact that features learned in lower layers are more general and thus more transferable in comparison with the features of the higher layers which are known to specialize [15]. This observation implies that models which hope to be effective at continual learning need to be wider in the higher layers to accommodate for this lack of transferability of the features at these scales.

## 5 Conclusion

In this work we have introduced an intuitive lifelong learning method which leverages the over-parametrization of neural networks to train new tasks in the inactive neurons/filters of the network without suffering any catastrophic forgetting in the previously trained tasks. We implemented a controlled way of graceful forgetting by sacrificing some accuracy at the end of the training of each task in order to regain network capacity for training new tasks. We showed empirically that this method leads to results which exceed or match the current state-of-the-art while being less computationally intensive. Because of this, we can employ larger models than otherwise possible, given fixed computational resources.

Our methodology comes with simple diagnostics based on the number of free neurons left for the training of new tasks. Model capacity usage graphs are informative regarding the transferability and sufficiency of the features of different layers. Using such graphs, we have verified the notion that the features learned in earlier layers are more transferable. We can leverage these diagnostic tools to pinpoint any layers that run out of capacity prematurely, and resolve these bottlenecks in the network by increasing the number of neurons in these layers when moving on to the next task. In this way, our method can expand to accommodate more tasks and compensate for sub-optimal network width choices.

*Acknowledgments* We would like to thank Kyle Cranmer and Johann Brehmer, Dmitri Chklovskii and Anirvan Sengupta for interesting discussions and input. SG is partly supported by the James Arthur Postdoctoral Fellowship. MK is supported by the US Department of Energy (DOE) under grant DE-AC02-76SF00515 and by the SLAC Panofsky Fellowship. This work was partly supported by NVidia (Project: "NVIDIA - NYU Autonomous Driving Collaboration").

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
