# OpenReview forum: "Continual Learning via Neural Pruning"
_NeurIPS.cc/2019/Workshop/Neuro_AI — Real Neurons & Hidden Units @ NeurIPS 2019 Poster_

### Official Review · AnonReviewer2 · 2019-09-26
**Interesting, albeit straightforward approach to minimizing interference**

**Clarity:** 4

**Category:**

AI->Neuro

**Clarity Comment:**

The paper is well-written. However, additional discussion about the central assumption of the model, that the "interference" weights can be set to zero and ignored, would be helpful.

**Evaluation:**

3: Good

**Importance:**

3: Important

**Importance Comment:**

The authors use sparsification to study continual learning. They claim this is superior to previous approaches that expand networks for subsequent tasks or penalize changes in previous weights. That being said, I am not convinced that this approach is really that different from previous approaches that expand network size with new tasks, since the authors are essentially forcing each task to use largely nonoverlapping subsets of the network

**Intersection:**

3: Medium

**Intersection Comment:**

The authors attempt to connect their results to neuroscience by noting the plausibility of their approach. However, the results seem to suggest a sparsening of representations from lower to higher layers in the network, which at least for the visual system seems it may be counter to the experimental findings. Also, there is no discussion of the biological process corresponding to the determination of which weights are "interference" weights during the learning of a new task.

**Rigor Comment:**

The authors compare their results on permuted MNIST and split CIFAR. For the latter, the results are compared only to Zenke et al. 2017. It would have been nice to see a comparison to a network with non-fixed architecture but comparable network size after training on all tasks.

**Technical Rigor:**

3: Convincing

---

### Official Review · AnonReviewer3 · 2019-09-26
**Interesting proposal to do continual learning by pruning with preliminary promising results**

**Clarity:** 3

**Comment:**

The paper proposes a new method to perform lifelong learning. The basic idea is to prune the neurons of zero or low activity and use these neurons for later tasks.  The pruning procedure leads to a set of weights which could be changed freely without causing any change to the output of the network.
I have not been following all the previous work on continual learning. But I really like the idea and the approach the authors are taking. The results shown in Fig. 3 are promising.  Overall, I think this is a strong submission.


**Category:**

Common question to both AI & Neuro

**Clarity Comment:**

I found the writing is generally clear. It is not difficult to follow the paper.

**Evaluation:**

3: Good

**Importance:**

4: Very important

**Importance Comment:**

This paper attempts to address an important problem. The method proposed are intuitive and reasonable, which could potentially inspire future work.

**Intersection:**

3: Medium

**Intersection Comment:**

The paper would be stronger if the authors could refer to some neuroscience literature on pruning of synapse in the brain.

**Rigor Comment:**

- The authors tested the method in a two sets of experiments. The task is created based on permutation/split of images, thus the tasks are quite similar. Did the authors tested quite different tasks, for example, learning to classify MNIST then CIFAR and so on?
- In terms of parameter m, the authors used 0.05%-2%. Would these numbers generalize to new tasks?


**Technical Rigor:**

3: Convincing

---

### Official Review · AnonReviewer1 · 2019-09-27
**Impressive step forward**

**Clarity:** 4

**Comment:**

It would be great to back up these empirical findings with some mathematical analysis, even on a toy version of the model. The idea makes intuitive sense, but fully exploiting it and indeed understanding its limitations is going to be hard to do with experiments alone. This may for example help with principled selection of the hyper parameters depending on the data structure.

**Category:**

Neuro->AI

**Clarity Comment:**

This is an excellently written paper, carefully covering the background literature, well-paced intuitive explanation of the key idea, and straightforward presentation of the results.

**Evaluation:**

5: Excellent

**Importance:**

4: Very important

**Importance Comment:**

This is a clever idea, implemented well, and showing good progress on an extremely difficult and important problem.

**Intersection:**

3: Medium

**Intersection Comment:**

The innovations are biologically inspired, but it is clearly an ML paper. It is not obvious to me that the findings have any direct implications for our understanding of the brain.

**Rigor Comment:**

The methodology and analysis are as rigorous as field standards. I might have liked to see plots of the validation performance as a function of the three hyper parameters optimised using grid search, to get a feeling for the robustness of the methods (the plot in Fig 3a implies that the results are quite sensitive to these choices).

**Technical Rigor:**

4: Very convincing

---

### Decision · Program_Chairs · 2019-10-02

Accept (Poster)